# The Co-Construction of an Elegant Ending—Polyphonic Musical Intervention in Palliative Care: A Case Study

**DOI:** 10.3390/bs12100392

**Published:** 2022-10-14

**Authors:** Raffaele Schiavo

**Affiliations:** Art Research Education, Hospice Palliative Care Unit, Rizza Hospital, ASP 8 (National Health Service), 96100 Siracusa, Italy; raffschiavo@gmail.com

**Keywords:** polyphony, helping relationships, complexity, social body, palliative care, music education, human voice, civil behavior, end of life, aesthetic criterion

## Abstract

The complexity of creative processes, within which human nature pines and finds itself, reflects a state of relational emergency. The weak structure of our multicultural system gives us a series of behavioral flaws, denouncing our inability to welcome diversity and treasure it. In the search for the way out, using the metaphor of polyphony is increasingly frequent. However, since the term Polyphony is borrowed from musical language, it would be necessary to experience it concretely, and this usually does not happen. Those musicians who aspire to be *artists in helping relationships* should constantly train themselves in this sense and force themselves to make their ability understood externally, helping those who for reasons of life find themselves experiencing the limit. The field of palliative care is probably the most suitable socio-cultural setting for getting in touch with the personal (mis)interpretations, idiosyncrasies, and pain of those who feel close to the end. To their aid, the aesthetic criterion advances. Making sacred the unifying experience of loss and finiteness turns into an educational process moreover therapeutic, in the co-construction of an elegant “finale” able to reach the heart and intelligence of those who remain.

## 1. Introduction

All those musicians who studied the art of composition must know how to create polyphonic narrative structures, be able to fit different sounds and individualities well together, design complicated rules of emotional–relational nature, and transfer their sophisticated intrigues of polyphonies on a map becoming a musical territory.

Clearly, those of them who are interested in helping relationships should be trained in understanding others’ difficulties first. If musical and interpersonal skills are required at the basis of such an education, it is also true that, on the other hand, a gradual overcoming of the sense of ridicule must be indispensable. Once the artists have acquired all the necessary elements to ensure proper conduct on their artistic stage, the practice of helping relationships shows its own stage as well.

The socio-musical artist’s creativity is subjected to a polyphony of purposes, which are obviously discordant by nature, and whose results, however, cannot betray the extraordinariness of coincidence, insight, and surprise. Sociologist Edgar Morin seems aligned with this leitmotif: “The problem of complexity is not to replace separability with inseparability, but inserting them one inside the other: connecting what is antagonistic or contradictory, since antagonism and contradiction appear as complementary [1]”. The recourse to the theories of complexity could facilitate the animated and intriguing rules belonging to the game of polyphonic composition, as well as the musical behaviors that derive from it and which could well be applied to a new behavioral level among people. Such a challenge requires processes of abstraction within which to consolidate the listening and the welcoming of contradictory thoughts and events. Additionally, all the paradoxes that existence mercilessly hands over to the world. Never like now, the need for a different thought would be appropriate. It must be a thought capable of taking up the challenge of the complexity of reality; that is, to grasp relationships, interactions, and mutual implications, multidimensional phenomena, and realities at the same time in solidarity and conflict (such as democracy itself, a system that feeds on antagonisms when it regulates itself) [2].

The aesthetic criterion asks for help and trust in this type of academic reflection, retrieving them from the most disparate fields of study. Sven Hroar Klempe believes that complexity is definitely embedded in musical polyphony: “it interacts with our ability to make complexity meaningful”[3]. The enjoyment of complexity in polyphonic music demonstrates our ability to make an almost unmanageable complexity meaningful. His conclusions are summarized in this direction: “mental complexity can be expressed through polyphonic music” [3]. As a reflection of his thought, a musical proposal arises spontaneously in the opposite direction: polyphonic musical thought can well embody the mental complexity inherent in every human being; since musical polyphony is an extraordinary behavioral product of human nature, paradoxically bizarre and evolved, contradictorily delusional and enlightened.

The polyphonic orientation guides the artists’ training in helping relationships. It predisposes them to listen to others and facilitates their interaction while maintaining its complexity. Here neuroscience comes to the rescue of the visionary artist. Daniel J. Siegel well embodies, and with extreme grace, this complicated thought.

Continuing to be amazed by human nature, we may also be lucky to come across the best that neuroscience can offer to the torments of the visionary artist. Daniel J. Siegel well embodies, and with extreme grace, this complicated trail of musical rests: “We don’t live in isolation from one another. Our mental lives are profoundly relational. The interactions we have with each other shape our mental world. Yet as any neuroscientist will tell you, the mind is shaped by the firing patterns in the brain. And so how can we reconcile this apparent paradox–that the mind is both embodied and relational? Shouldn’t something like “a mind” be located in one place, come from one source, be owned by one person? That’s the question we’ll address in illuminating the nature of mind. (…) you can state that the mind is both embodied in an internal physiological context and embedded in an external relational context. Embodied and embedded is the fundamental nature of mind” [4].

To emphasize his words with the necessary variations to the theme, living a life dedicated to creativity would mean immersing oneself in a way of being that is permeated with a profound sense of presence and emergency. “Awareness and consciousness are of great interest to a range of academicians, from neuroscientists to philosophers, and they are the subjects of literature and poetry. The experience of music and dance bring the rhythmic motion of the body in tune with our inner subjective awareness. The fundamental nature of our relationships is shaped by awareness: when we share something in awareness with another person, it changes the nature of that experience, alters the flow of information processing, and creates the closeness we feel with another person” [4].

The artist in helping relationships is trained for the same closeness and chooses to intervene in the scene of despair and non-acceptance. In this sense, the palliative care area becomes the ideal stage. The curtain opens and there are many seats in this theater of unsolved polyphonies. With the entry of Thanatology among the most recent currents of study, the ancient becomes present: “This science emerged in a historical context marked by intense social, economic, and political changes that contributed to the concept of death being excluded from social life” [5]. Who states this, is Ines Testoni from Padua University, a pioneer among the trainers of this sophisticated and inclusive discipline. In addition, she firmly believes that palliative care units and hospices are places of valuable learning: “The denial of death in Western society deprives young people of the tools to derive meaning from experiences of death and dying. Literature shows that death education may allow them to become familiar with this topic without causing negative effects” [6].

Here another compositional skill advances, which is improvisation: an extemporaneous quality that tests all the performers on stage. The context of the end of life allows us to start backward, from the construction of the Finale to get to retrace the progressive revision of everything that was behind, from the very first bar. Experimenting with it in learning and verifying workshops, whether it is really possible to move polyphonic thinking from one life scenario to another, towards musical interaction strategies to be taken in everyday life and among people, beyond music, beyond words.

## 2. Materials and Methods

The socio-musical method here presented combines elements of auto-ethnography, musicology, and behavioral philosophy, in an attempt to find the justest and most effective way to connect together all those disciplines projected to assist the person, considered in his/her full bio-psycho-social and spiritual reality.

The primary purpose of the method is the training of the *artists in helping relationships* so that they can move in an agile way from performance to therapy and thus intervening in the educational, artistic, rehabilitation, and health care field, with particular attention to the end of life ambit and in palliative care.

The name given to the method is VoxEchology [7], wanting to emphasize its three fundamental elements:(a)The human voice (Vox), to be understood as an acoustic extension of the body and precious instrument to be refined, both individually and in groups, for its extraordinary interchanging characteristics;(b)The reflection of the voice (Echo), as auditory refraction that embodies all the female expressive potential, is now incited to free oneself from the constraints of its own myth; hence, a revision of the symbology that relocates Echo and Narcissus in a different representation and with happy contents;(c)The application of the geometric-musical language (Logy) to helping relationships, so that they can be intertwined with both theatrical procedures and the study of polyphonic vocal literature itself. The VoxEchology method allows to know and practice relational complexity schemes, which, by virtue of the Polyphony’s principles, happily weave together every form of diversity, contrast, and alignment, always practicing new paths for the search for beauty and shared well-being.

Polyphony is a fabulous art of musical composition because, in this regard, it recognizes the full dignity of human nature, with all its most unmanageable nuances. This is because its organization is governed by specific rules of interpersonal conduct that do not discard and do not fear paradoxical and contradictory solutions. Getting in synchrony with diversity. This is what a polyphonic mindset can offer, when trained to be converted into action, with all the attendant risks and responsibilities. Neuroscience meets this reflection, which is already practical experimentation for those artists in helping relationships involved in their workshops. It is very interesting to understand how synchrony, from this assertion, therefore, is strictly connected to a mental intrusion of musical nature, even when it occurs in contexts where music itself had nothing to do with it. In any case, for a greater linguistic and socio-musicological carefulness, it would be better to speak not of “music”, in an abstract way, but of “musical language”; since it implies a dialogic formulation, ruled by grammars and syntaxes not so far, yet unrelated to spoken language. Khalil, Musacchia, and Iversen investigate and study the Interpersonal Neural Synchrony (INS) which, indirectly supporting the VoxEchology method, appears to be “correlated with increased behavioral synchrony during social interactions’ and may represent mechanisms that support them” [8]. They live thanks to this extraordinary mechanism of synchrony, that could be defined as compositional-musical.

Polyphonic thought guides the voices’ behavior in the field. First of all, entrusting in a vertical force able to align each voice together within a transitory obligation of understandings and agreements: it is called Harmony. However, before reaching these states of mutual consent, the polyphonic process achieves its splendor in the paradoxical interlocking among sequences of different voices, proceeding in a horizontal way and in clear contrast to/versus each other. This extraordinary compositional procedure is called Counterpoint and represents the true research core of the VoxEchology method. It is the cultural challenge for the construction of a new behavioral science: a unique experiential path, structured on a musical intelligence that is aggregating, orchestral more than choral, really capable of giving life to relational architectures, to be led by polyphonic ideas and actions.

A basic musical education founded on these principles requires that polyphonic thinking could be incorporated, embodied. Eva Vass, teacher and researcher in the vast field of pedagogies, argues that “to reimagine education, we need to return to the body” [9]. She achieves it by presenting a series of stimulating questions, including these two: “How can we capture the significance of embodiment in learning and knowledge building?” And thereafter: “If the body is the medium of collective experience, how can we tune it well?” [9]. A body training in musical polyphony would facilitate the development of affective connections that can be embodied and then incorporated in an elegant, graceful, and aesthetically effective way. These questions stimulate socio-musical research based on the body and the voice as its extension. In addition, wanting to play with the words of the biologist and educator Alan Rayner [10], such questions predispose the researcher to rely on that flow the space owns, when it enters into a relationship with the energetic form of its proper receptive qualities.

The interest in end-of-life and palliative care is part of this design [11]. When we begin to understand certain analogies with the layout of a whole life, even the revision or the creation of a piece of music becomes the performance upon which to build the sacredness of a lost rite, to be embodied and show the world. It is the reacquisition of what is strength, spirituality, integrity, and beauty for the social body. In its compositional scan of beginning–development–closure, such a musical story implicitly invites us to reconsider the personal conduct of each one in the score of a shared life, as if the characters of a community were voices of musical instruments, immersed in the composition of oneself with others.

The need to give the most worthy attention to the construction of an elegant ending, of a Finale, arises from this sociological drive, especially when the aesthetic criterion proves to be able to find its right application among the difficulties of health policy.

### Case Study (First Part)

Among the many stories to tell, one of the first in particular is close to my heart and it is Tina’s (pseudonym), about an elderly woman who loved poetry. She was very sensitive to music and theater. I met her for the first time in 2002. She was part of a cancer patients group involved in my music therapy program, organized, and co-conducted together with a competent psycho-oncological hospital team. Tina stood out for her collaborative skills. Her gesturality was extraordinary, as well as her voice and the sacredness that she adopted in a way of building empathic relationships amongst the participants. For her, every meeting was a ritual. She definitely had a proper style in manipulating certain sound objects that we used those times as musical instruments within our psycho-oncology group: metal plates on wooden cubes and some sort of different percussions, all easy to be played for creating musical dialogues.

I gave these curious instruments to the group’s elements during the meetings. Tina’s own way of playing was full of theatricality and simulations, for a magical world that through her performances became lively and present. She was diagnosed with cancer when she was still mourning the death of her son, stolen from life by the same disease. The woman had no scruples in presenting her virtuous and extravagant relationship with her son’s image. Those musical dialogues with the invisible kept the memory of that sacred affection alive. Gestures and sounds were converted into structures of gentle, elegant, and alternative conversation with us.

Her improvised performances were animated by a sort of candor. She immediately felt the daily mood emerging from that special tribe. Just as she knew to be aware of supporting me very well in her role of cautious assistant and collaborator. She indulged in eccentric prayers and rituals without clear verbal explanations. The intensity of the movements and the quality of the vocal sounds reached her to something like a state of ecstasy, where getting lost together with her creative imagination, inside a sea of light covered with tears and joy, confident of our support. Nobody dared to criticize such a performance, full of dignity and beauty.

I remember when the program was over, Tina followed me in various shows and city conferences. At a certain point, I became aware of her absence. I did not see her for quite a while. Several years later, I accidentally met her, after a yoga path she attended. Visibly worn out, but smiling. She felt pain from a new chemotherapy cycle. It was the last time I saw her around.

In the spring of 2012, I was contacted by the palliative care unit. They let me know about her terminal state and about the home care program already started. Tina wanted to see me and play together again, this time in the company of her daughter Una (Pseudonym) [12].

The VoxEchology method follows precise aims, through which palliative care could be redesigned:-A fundamental aim is to address the complexity of human relationships so that it is not fought or demolished, but re-contextualized in the name of the aesthetic criterion; that is, acquiring intuitive richness and disciplined curiosity in the search for what is beautiful.-To create the missing link between the humanities and the scientific disciplines, lightening and thinning their borders within a safe transit area, where the fluid interchangeability of each person’s features could be really allowed, together with their own bio-psycho-social and spiritual dignity.-To share the idea of a different musical education, whose didactic-training energies are not dispersed over easy talents and obvious emotional responses, but committed to experimenting with new patterns of social behavior, holding the principles of Polyphony as a primary reference.-To facilitate the construction of a polyphonic awareness, especially where the scanning of creative cycles between individual and group identities is particularly difficult.-To create a code of social conduct that is capable of guiding people towards a logic of happy joints. A language, where the musical polyphony of conflicts and agreements becomes a theater of beauty; where paradoxes and contradictions are necessary for the realization of new frames of meaning; and where the interlocking between nature and culture is not feared, but gently provoked for a wider aesthetic elaboration.-To conceive a program of artistic studies, the application of which is focused on extraordinary work contexts, like palliative care–end of life could be. This is how the interest in broadening the sense of interdisciplinary knowledge could emerge, as well as the artistic possibility of intervening in issues of social importance, concrete and to be embodied.

For the aims set out, some general goals follow. Most of them correspond to the precise features that make this method distinctive.

-To track down, acquire, and show artistic qualities and musical–theatrical skills, able to be translated into models of helping relationships.-To present Musical Polyphony as a form of relational architecture, so that the concepts of performance and ensemble music are reinterpreted in a sociological–therapeutic key.-To rethink the entire basic programming for musical studies (e.g., theory and solfeggio, instrumental techniques, history and analysis, ensemble music) for a game of metaphorical practices that brings out hypotheses of aesthetic parallelism between the construction of polyphonic thought and experiences of civil conduct.-To set up group experiential workshops (interactive lessons distributed as training, exhibition, and verification levels) within which to elaborate the aims and goals here presented. Repertoires of ancient music (from the Middle Ages to the Baroque), techniques of vocal dyplophonies (Overtones Singing), and some original contemporary musical styles (with a careful eye towards the minimalist approach) will be studied. From their compositional structure can come valuable suggestions for building models of helping relationships based on aesthetic–musical principles.-To define a training program focused on the conquest of an intra/interpersonal awareness based on musical exercises and games, aimed at the successful outcome of a performance and at a continuous regulation of one’s emotions, in the implementation of sophisticated relational strategies.-To think and build artistic events for therapeutic purposes, reinventing an aesthetic of performance that fully involves the performers, the composers, and the public.

The method also includes some specific goals, in this case with respect to palliative care and the end of life:-To redesign palliative care along the interaction musical principles of Polyphony, considering the VoxEchology method as a strong example of this redesign.-To help patients and their families to recover aesthetic relational modalities, leading them towards new qualities of knowing, feeling, and building together.-To encourage the sense of social cohesion in a creative perception.-To promote musical intelligence towards opening relational mental states and making possible the aesthetic contextualization of pain, the irreducible symptoms, and the sense of imminent loss.-To create a privileged context in which to pour out anger and pain, sense and meaning, tension and relaxation, relief and peace.-To facilitate the work of all professionals involved in the home care program, by creating for them a special training activities platform, to be combined with their updating courses.

## 3. Results

### Case Study (Second Part)

Tina was easy to sing. So already at the first meeting in her house, I saw her rapidly starting with some vocalization. Her hands tried to intertwine mine in a dance of sighs. I could not help but observe a drip attached to her left arm. It was certainly not physiological liquid, but a dose of chemotherapy; yet another and last attempt to bring her back to a distant past, in a forgotten state of health. The husband and the other son did not accept the poor prognosis. So, they wanted to join a further drug assistance plan, encouraged by a team of doctors different from ours, unfortunately acting in parallel autonomy. We provided free palliative care at home, the other team proposed vain hopes. Musically speaking, this was not a counterpoint at all, but the performance of two different pieces of music, totally different and uncoordinated between them within the same contest of listening. This contrast was unfortunately visible, moreover testified by the important absence of the two men during our meetings. Although starting from the third meeting, sometimes we timidly observed a chink opening from the door: we were able to catch an interested, curious, respectful face of a male. However, neither of those two men ever took part. Anyway, we set up a calendar of meetings, establishing all those modes of action that could be useful for our performance together.

The most used one was the VoxEchology improvisation sequence: that is, a sequence of thoughts and sound–gestural actions closely linked to the socio-musical method presented here. Figure 1 shows in the center and vertically, proceeding from top to bottom, the directions to follow during the overall improvisation. On both sides, two series of different phases are shown in parallel, which reflect the same central procedure, however using other terms. On the other hand, Figure 2 shows the same sequence of indications always arranged in the center, but on a horizontal level. The reflections wander in parallel, upwards as well as downwards, ideally suspended for a possible hypothesis of randomness in a three-dimensional way, which gives the performance’s practice the opportunity to preserve memory of what happened or could happen during the performance.

The elderly lady was enthusiastic about the presence of her daughter Una and what a magnificent trio was ours. I asked her if I could bring the video camera. Both women agreed. They were ready to give the world their testimony of such an extraordinary adventure. Lying on the bed, drawing shapes in the air with her arms, Tina sang in a voice that hardly sounded like hers. Sometimes she could even lift her feet, making them dance. Building rituals of authentic movements while vocalizing, she repeated them from meeting to meeting, with a certain theatrical and choreographic discipline. Until the ending came together, even when she was very tired. Her songs and her voice’s timbre changed further during those altered states. Tina assumed a musical profile very similar to that of an Indian–Pakistani shaman: musically speaking, those traits were all there (Figure 3). Her daughter Una and I played and sang with caution and so we proceeded with our expressions towards her. It was all fantastic and how lucky to have taken it back in videos.

Before continuing, it is indispensable to underline the questions concerning the ethical parameters. This study was conducted according to the guidelines of the Declaration of Helsinki. Ethical review and approval were waived for this study, because the Ethics Committee within the National Health Service for the Rizza Hospital, Palliative Care Unit-Hospice Kairòs (ASP 8 Syracuse, Italy), with regard to case reports, considers necessary only the informed consent of the patient or, if the latter is deceased, of their family members. Furthermore, about the video presented on Youtube in 2012, it was correctly posted on the internet as well. That was the official and legit praxis, institutionally authorized by the Italian Society of Palliative care (SICP) for inviting presenters and lectures involved in the SICP annual conference (Turin 2012), in order to participate in its video contest about palliative care and end of life themes. Here is available the document that the hospital team usually presents to patients and familiars as acceptance of recovery and caring plan, also in asking to agree or not agree to the opportunity of showing data, materials of all sorts, audio and video demonstrations of a special program such as this and so on.

She often brought out the same theme on different days, during her sound–gestural trance states. Most of those sequences became thematic, obstinate, and rigorously ceremonial. No one can explain where that strength of choreographic energy and musical insights could come from. Then, the continuous rotations of her right arm, the one free from the drip, and both arms tense and bent. She drew an oval perimeter around her body and up along the air. She brought that hand to her forehead, with the palm facing outwards. It was as if she were raising a weight, or perhaps radiating a light upwards, extending both the fingers of the hand and then the whole arm. All of this, with great resistance for at least half an hour. Her trance revived her face, staring at an expression that well masked the serious powerful sounds of her surprising vocal timbre: a sound confident though hoarse. All sounded so archaic as if she had recovered something lost: something not belonging only to her, but to all of us present there. Once again, Tina came into contact with the invisible, in an extraordinary blend of reflections with a diverse, generous version of herself.

According to artistic parameters, the relationship built with her daughter Una was very tender. Seeing them relating in that form, so foreign to their usual daily life, was an intense and penetrating show. How many times we ended up laughing, all three together like kids. Tina would never have expected such strong participation by Una in all our musical games. Seeing her daughter placed on the bed, dancing on her knees and singing next to us, reminded me of a series of theatrical and musical images to which I always tried to respond adequately. Their sense of the ridiculous gradually diminished, while mine was trained. Each meeting had become a pretext to bring out old and new gesturality, melodies, and shapes that were accumulating in our minds. However, we also set aside time for studying songs from my medieval sacred repertoire.

As the woman definitely progressed towards her stillness, her relationship with me was reducing more and more. Instead, their mother–daughter unity emerged thick and rich (Figure 4a,b). They had perfectly entered the rules of a new game, choosing to adopt those rules, even during the incursion of the most unpleasant and unsustainable symptoms. Even handing the basin to the mother, when she was in the throes of vomiting, while I was supporting her with singing and gracious theatrical behavior. Their bodies were busy working differently together. The mother responded to everything that came from her daughter, in this extravagant relationship, made of art, love, and mutual help. I felt I was no longer needed. I understood that I had to withdraw and stay listening.

They asked me to attend the funeral. So, I intervened in the church over the rite, using the same songs we improvised at home and a few more songs taken from my medieval repertoire. The grieving community was urged to sing and rejoice in Tina’s life, lived and concluded in the fullness of her great Finale. After some hesitation, the whole assembly began to sing, sometimes even smiling with me and Una. Never in detail, had I seen the corpse of a person so well before, even more, the body of a loved one. It was the first time. I saw a wax doll, an artifact, an empty container, a human figure emptied of life and embodying the vague appearance of someone I seemed to recognize. Somewhat infinitely dear and powerful had flown away. Shortly afterward, I would have seen lifeless bodies in large numbers and varieties. I never imagined I could stand next to them, dancing and singing with such naturalness. As if I had always done it. As if someone had taught me that, a lifetime ago.

## 4. Discussion

The therapeutic outcomes of this particular socio-musical method of intervention can be seen ongoing both among patients and among all the people involved, also outside the care program (family members and communities to which they belong). There is always an extraordinary connection between inside and outside, between the individual and the social body, between a sense of the species and a limited perception of oneself. Maurice Merleau-Ponty writes: “Loneliness and communication must not be the two terms of an alternative, but two moments of a single phenomenon, since in fact the other exists for me” [13].

From the stories of patients and family members, the community feels to be involved in a certain kind of collective awakening, as it happened during the direct experience of the funeral: a form of activation among people gradually perceiving the penetrating force of the group in action, almost an aesthetic–political necessity of being and building together.

In the specific case here analyzed, Una, Mrs. Tina’s daughter, had felt a growing transformation within herself, new information about the end and the Finale. She herself felt different, willing to live within territories of behavior never experienced before. Although she knew about different artistic and cultural activities in which her mother used to participate, it had never crossed her mind to be able to share them. The strong affective-emotional drive, perceived as approaching the final separation, will probably have increased her need for sharing and contacting with a well-organized aesthetic reality, paradoxically liberating and nourishing. There are writings and videos attesting to such events.

With respect to the benefits, the conditions of analysis and test could take on a different formality.

A first essential benefit here consists in giving due importance to the relational opportunity provided to patients. An assistance and accompaniment program following educational and artistic orientation is able to create a framework of meaning, connecting the patient’s interpersonal world together with all the variables of complexity that a life spent with others would entail—and his/her most extravagant intimacies: that is, of that internal and little revealed world, which can be defined as intrapersonal. Here the continuous questioning, the complicated succession of dialogues with the most diverse Selves and for which the person is distorted, must not be able to affect psychiatric pathology.

On the contrary, a highly empathic intelligence may be able to understand the dramatic context within which terminally ill patients and their loved ones get lost and despair. This degree of empathy is well rooted in the artistic–relational qualities of those who are able to welcome all manifestations of the unreasonable with aesthetic references. In itself, it is a source of benefit for the patient, who in return warns of the full satisfaction of the agreement with the artist.

It is necessary to be aware of how, faced with the prospect of death, patients cannot help but be disoriented within extreme creativity, to the point of touching delirium and disruption. Here comes the benefit of feeling understood and welcomed, so it becomes possible to receive sensible answers, although according to artistic criteria. It is the benefit of feeling the orchestrated sounds of pain’s shocks and devastation, that no one would ever have the courage to take as a model to reconstruct the scene of discomfort.

The artistic expression of finding oneself building together is experienced as a benefit, a relief, and a gift. The musical structures conceived and improvised by mutual agreement become incomparable opportunities for mutual satisfaction. Within such a performance, there is even room for the well-being of those people in the health care team who, in one way or another, wish to change their clothes and free themselves on the stage for singing and dancing in a group. Those times would coincide with the possibility of subconsciously providing new and unexpected benefits.

During the artistic–therapeutic encounters, there were also forms of temporary resolution of the most common unpleasant symptoms (essentially some breathing difficulties, movements, and elaborate reflections), probably stimulated by a different contextualization of their own diseases and emotional suspensions. The attempt to keep the mental abilities of patients active through the study of ancient songs, both sacred and profane, generated in them a notable sense of self-esteem and personal gratification: feelings dictated by discovering and recovering a culture lost in the meanders of monotonous habit and in a weak desire for knowledge.

The ritual involves completing the work of each extraordinary rapprochement with the dying person.

The reduction of unpleasant symptoms, psychological and philosophical work, direct and indirect, on one’s own conditions and the surrounding context, contribute to the creation of new frames of sense and meaning. Each event can be artistically translated into the here and now of the therapeutic relationship.

In the case of terminal patients and their dear ones, all there on the scene, it is necessary to be ready for returning what they desire from the artist, in the creation of a path of creative cohesion, characterized by mutual knowledge and relational awareness, through a language that includes in itself all the elements of the empathic approach.

Any form of narration, of artistic rendering of those extraordinary events carried out with patients and with their most disparate guests, could represent a different way of collecting data, in aid of broader socio-musical research. The video material showing some events relating to the context exposed here represents precious data, as well as opportunities for study and insights. Some narratives have been transcribed in different publications in the Italian language and some presented in English for palliative care congresses abroad [14]. Although they are not part of a rigorous scientific formality, they all remain valuable evidence of happy relational interlocking, in the resolution of anguish and hurt, as well as the risk of insanity and loss of identity on the part of patients.

With respect to the conductor’s ability, it is obviously important to know how to manage and regulate his/her emotions during the overall performance. The artist in helping relationships must be trained to contain every psycho-organic intrusion; and therefore, anything that could compromise his/her own actions and presence on the scene. As a well-trained musician–singer, he/she will be able to combine together how to make and how to listen to music: two practical skills that must be fundamental for the expert musician–theatrical performer and that cannot proceed separately, but always together and in total extemporaneousness.

In any case, whatever specific environment the artist should visit, hospital or theater, the mask of fiction and the simulated alteration must be supported by a specific control of sounds and gestures. The immediacy of the aesthetic criterion adopted must be planned after constant exercises with others and with direct experience in the field. Especially, when the artistic intervention comes to coincide with the delirium of the patients, who, sedated, upset and in pain, rely on what the conductor creates artistically together for them and with them. That could be: the illusion of having a sacred frame of meaning, the opening towards a new alliance with the unknown, and the artistic relief that will keep their dignity always alive. This is the distinctive, unique value of the VoxEchology model, compared with traditional palliative care.

## 5. Conclusions

In this type of path, it is far too simplistic to speak of music therapy, although the most suitable term already recognized in the sector literature could coincide with integrated music therapy and moreover with Creative Arts Therapy (CATs), a new field of research, where music, body, improvisation, drama, and arts of all sort are all interlaced together [15]. In fact, there is also a lot of experimental teaching and enough socio-psycho-pedagogical material.

At the end of this articulated exposition, some specific questions about the VoxEchology method guidelines could better enforce its extra-academic potential, hopefully to be converted in the future as an institutional path of study within a stimulating interdisciplinary contest:

What would human relationships look like, if it were possible to apply the principles of musical polyphony to individuals’ behavior within their conflictual everyday life?

Would it be enough to learn the art of musical polyphony within appropriate experiential laboratories (workshops), distributed by levels and learning times, with the necessary demonstrations and verification, in such a way as to be able “to contaminate” their intra/interpersonal dynamics?

In a performance conceived as a relational story between performers and audience, following the principles of polyphony and continuing to use the strategies of theatrical fiction, what novelties on the level of civil reflection could emerge?

-How could ideas on helping relationships vary, if the impulse to rescue people were governed by patterns of musical interaction?-How much would the human instinct change, once the elements of polyphonic thought were acquired and consolidated, with respect to the meaning of life and death in their cyclic unity?

These questions draw a map of the learning units, about verification, evaluation, and monitoring processes, based on the practical experience of Research-Action. It must proceed with corrections and adjustments of the musical behavior, individually and in an ensemble. It follows the direct elaboration of the principles presented up to now, as well as their appropriate distribution within the most varied institutional containers, appropriately charged with forming “artistically” future citizens and responsible performers [16].

These questions draw a work grid that is certainly unusual for the creation of new learning units, also with respect to the verification, evaluation, and monitoring processes that would derive from it. In any case, on the basis of practical experience in Research-Action, it is already possible to proceed with corrections and adjustments to one’s behavior on the stage, as in some small and significant traits of social life [17].

There are now many artists helping relationships around the world. They are cleverly networked and all of them are working together, committed to building future citizens with a substantial aesthetic sensitivity. They face the unmanageable complexities of human nature on a daily basis because they trust in a solution that can arrive, sooner or later, at their destination from the happy and supportive encounter between art and science.

As regards future implications, the entire city community should be educated on accompanying the dying persons and on how to proceed in helping their family members. People should also have the opportunity to follow the training of medical assistants and socio-psycho-pedagogical staff in places of discomfort and pain. At least, being aware of what a hospice can even represent, because there is a lot of ignorance and necessity to escape from this uncomfortable topic, from this taboo, which is death. The area of palliative care can become a largely cultural dimension, since the question of death affects everyone, starting with the entire community to which people belong. Artists, in helping relationships, can have this specific role of welcoming, education, and training within a special middle-earth, branded by the adoption of the aesthetic criterion [18].

## Figures and Tables

**Figure 1 behavsci-12-00392-f001:**
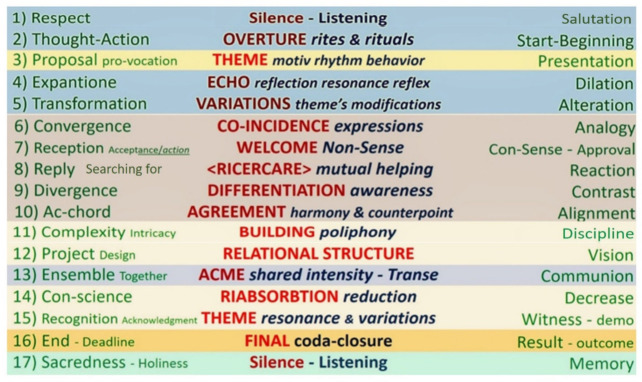
Here located in the middle, the sequence of musical actions which indicates the directions to follow during the overall improvisation. It proceeds vertically, from the first conduct Silence–Listening on top, towards the second one at the bottom. Both sides in parallel, on the right and on the left, show two series of different phases. Step by step, they reflect the same central procedure, however using different terms, essentially coming from diverse points of perception.

**Figure 2 behavsci-12-00392-f002:**
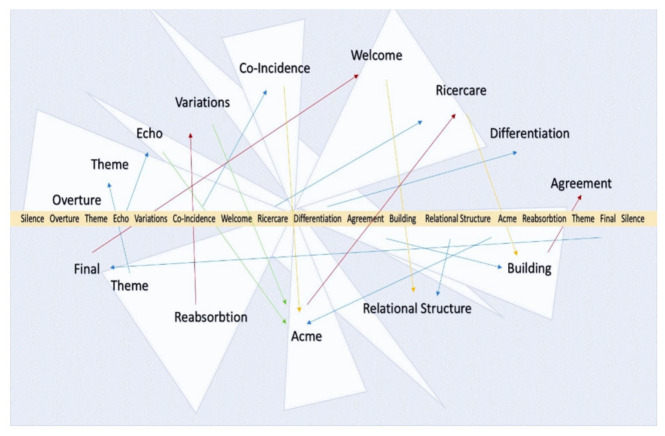
Here the same sequence is indicated as in the previous picture. It is always located in the center but follows a horizontal plane: a different way of showing the same sequence and its reflections that correspond to different levels of perception. The reflections wander in parallel, upwards as well as downwards, ideally suspended for a possible hypothesis of randomness in a three-dimensional way, which gives the performance’s practice the opportunity to preserve the memory of what happened or could happen during the performance. Those arrows in the picture point toward different possibilities to create associations of perception in regard to the different memories of every single moment of expression, corresponding to the different levels of the sequence to be played together, between the different participants on the same precise scene. It is something to be seen as multidimensional, compared to the bi-dimensional one which is Figure 1.

**Figure 3 behavsci-12-00392-f003:**
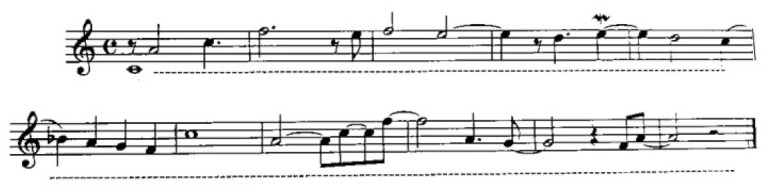
This is the music score of what Tina sang repeatedly, but with some small variations.

**Figure 4 behavsci-12-00392-f004:**
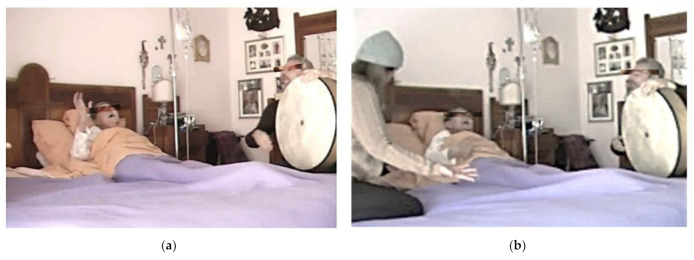
(**a**) In this picture, you can see Tina on the left, the patient, in her bad playing and singing with the conductor Raffaele Schiavo on the right. (**b**) This other picture, compared to previous one, includes the presence of Tina’s daughter on the left, as new guest performer on the scene.

## Data Availability

Not applicable.

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
