# Peer review of "The Co-Construction of an Elegant Ending—Polyphonic Musical Intervention in Palliative Care: A Case Study"

_behavsci, 2022, doi:10.3390/bs12100392_

Round 1
Reviewer 1 Report
Summary
This manuscript presents a poetric auto-ethnograpic account of an innovative music/voice-based model for palliative care, called VoxEchology. The aim of this model is the betterment of end-of-life experiences of patients in palliative care, building on core concepts and principles of musicology.
Conceptually and philosophically, this model is breaking grounds by bringing in parallels between artistic/musical polyphony and improvisation and the polyphonic/improvisational artistry advocated for palliative care.
Whilst the foundations for the vision of palliative care (VoxEchology) are rooted in musicology, there is ample evidence of transdisciplinary significance, indicating that the musical principles and metaphors (musical polyphony, relationality, improvisational artistry) help locate the masked, disregarded, little-understood yet foundational elements of the human self - or nature itself - which in turn lead way to creative co-therapeutic (or therapeutic-educational) practices in palliative care. All this points towards the need for (therapeutic as well as educational) co-construction of an ‘elegant ending’, just like a musical piece.
My main concern is that I cannot evaluate the manuscript on traditional scientific criteria, as it is not written in the traditional scientific genre. It would be better considered as an autoetnography - experiences of the therapist that lead to the vision and model of VoxEchology (or put the philosophical foundations of this model into practice).
I would be happy to review a significantly revised manuscript that follows the scientific tradition. For this, the key review questions set by Behavioural Sciences would be useful:
- Is the manuscript clear, relevant for the field and presented in a well-structured manner?
- Are the cited references mostly recent publications (within the last 5 years) and relevant? Does it include an excessive number of self-citations?
- Is the manuscript scientifically sound and is the experimental design appropriate to test the hypothesis?
- Are the manuscript’s results reproducible based on the details given in the methods section?
- Are the figures/tables/images/schemes appropriate? Do they properly show the data? Are they easy to interpret and understand? Is the data interpreted appropriately and consistently throughout the manuscript? Please include details regarding the statistical analysis or data acquired from specific databases.
- Are the conclusions consistent with the evidence and arguments presented?
- Please evaluate the ethics statements and data availability statements to ensure they are adequate.
However, I see this author's work to be different from a traditional empirical study. I can see two main directions, equally valuable:
- the presentation of the content as an auto-ethnography (either in a journal article or a book) that details the development and practice of VoxEchology as an innovative palliative care model. [For this, there needs to be a more detailed discussion of the ethical side of this auto-ethnography, following the core principles of ethics in social sciences).
- The presentation of the philosophical/conceptual underpinnings of VoxEchology, in the form of an opinion piece or a philosophical article. This would be especially valuable if the aim of the author is to rethink, redesign and re-imagine end-of-life care as a 'creative co-construction of an elegant end'. This is a wonderful proposition.
I am uncertain whether either of these propositions would suit the context of Behavioural science - there may be a need to find an alternative platform for the worthy, novel and rich practical and philosophical insights presented here.
I am attaching the pdf document of the manuscript with my annotations.

Reviewer 2 Report
It's in interesting article. The introduction is quite philosophical, while I expected to read some more about lifespan psychology/ the end of life and socio-cultural issues. However, this article focuses on one specific case in one specific setting.
In the materials and methods section, I miss a description of the research methodology. Was there a specific research question? What research design did you use? How did you analyze the data. You convinced me without doubt of the value of the intervention you presented.
The description of PIC2 is not clear to me. I don't understand how to read it. Sorry.
